# Quiescence of Human Monocytes after Affinity Purification: A Novel Method Apt for Monocyte Stimulation Assays

**DOI:** 10.3390/biom12030395

**Published:** 2022-03-03

**Authors:** Minh-Thu Nguyen, Leonhard Hubert Schellerhoff, Silke Niemann, Frieder Schaumburg, Mathias Herrmann

**Affiliations:** Institute of Medical Microbiology, University Hospital Münster, 48149 Münster, Germany; leo.schellerhoff@uni-muenster.de (L.H.S.); silke.niemann@ukmuenster.de (S.N.); frieder.schaumburg@ukmuenster.de (F.S.); mathias.herrmann@uni-muenster.de (M.H.)

**Keywords:** monocytes, human blood, CD14 magnetic bead, stimulation, bacterial lipoproteins, cytokines, repellent surface plate

## Abstract

Several methods to isolate monocytes from whole blood have been previously published, with different advantages and disadvantages. For the purpose of cytokine release assessment upon external stimulation, the use of monocyte preparations consisting of non-activated cells is prerequisite. Affinity-isolated monocyte preparations from peripheral blood mononuclear cells (PBMCs), obtained via positive or negative selection using magnetic beads, released pro-inflammatory cytokines such as TNF-α and IL-6 even without adding external stimuli, hindering any assessment of an effect of bacterial lipoproteins on cell stimulation. Hence, the cell preparation protocol was modified by adding a quiescence step on repellent surface culture plates, dampening any monocyte pre-activation. This protocol now provides a robust method to prepare silent yet fully activatable, pure monocyte populations for further use in stimulus-elicited activation experiments.

## 1. Introduction

Amounting to a proportion of approximately 10% of all blood leukocytes, monocytes play multiple roles in immune function, such as phagocytosis, antigen presentation, or chemokine and cytokine secretion [1]. It has been shown that monocytes confer an important role in pathogen clearance; moreover, they are relevant contributors to the inflammatory reaction as a consequence of infection [2]. Originating from bone marrow, monocytes circulate for a few days in peripheral blood before they migrate into tissues in order to differentiate further into macrophages or dendritic cells [3,4]. Human monocytes express cell-surface markers, i.e., CD14 or CD16, which have been used to categorize them into three subsets, such as classical CD14^++^CD16^−^ cells, intermediate CD14^++^CD16^+^, and non-classical CD14^+^CD16^++^ monocytes [4]. Among these categories, classical monocytes make up the majority; they are described as inflammatory monocytes with the potential for further differentiation into dendritic cells or macrophages [5]. In vitro assays analysing the pro-inflammatory reaction of monocytes rely on the availability of pure yet unstimulated monocyte preparations, a technical prerequisite that frequently challenges investigators.

Traditionally, the purification of monocytes from blood is carried out in two subsequent steps: (i) peripheral blood mononuclear cells (PBMCs) isolation and (ii) the isolation of human monocytes from PBMCs. In the first step, PBMCs are separated from other components of the blood such as blood plasma, erythrocytes, and granulocytes via density gradient centrifugation using Ficoll-Paque [6,7], or optional Hypaque-Ficoll for the further separation of mononuclear and polymorphonuclear cells [8]. In the second step, in order to isolate monocytes from lymphocytes, several methods are commonly employed. Historically, for instance, adherence [9,10] or cold aggregation techniques [11,12] were used. Another inexpensive method is the use of density gradient centrifugation employing hyperosmotic Percoll [13]. With the development of magnetic cell isolation technology, monocyte labelling with magnetic beads conjugated to mouse monoclonal anti-human CD14 antibodies allows one to effectively retain the monocyte population on a magnetic column while unlabelled cells are released with the void fraction. This method has been also designated as positive selection. Such CD14^+^ positive, isolated monocytes were then used for the in vitro generation of dendritic cells [14] or macrophages [15,16], or for studies on cell cytotoxicity [17] and migration [18]. Another method, also based on magnetic bead technology, was designated as pan monocyte selection, indicating the principle of the depletion of whole blood from all other cells, including erythrocytes, by magnetic labelling with antibodies against all blood cell surface antigens, and not being expressed by human monocytes.

In Gram-positive bacteria, lipoproteins (Lpp) play the main role in host innate immune activation via the TLR2-MyD88 pathway [19]. Previous studies have shown that monocytes are crucial for synovitis and joint destruction mediated by Lpp [20], or skin inflammation via TLR2 activated by Lpp [21]. These observations stirred our interest to investigate the interaction of bacterial Lpp and monocytes on a cellular level. In our model, primary monocytes were isolated directly from healthy human blood and incubated with synthetic lipopeptides which mimic different structures of bacterial Lpp.

However, similar to other researchers, we faced problems with monocyte activation upon purification. In this paper, we now demonstrate a robust and straightforward method for monocyte isolation precluding any cell activation prior to exposure with pro-inflammatory stimuli.

## 2. Materials and Methods

### 2.1. Ethic Statement

The use of human blood from healthy volunteers was approved by the Ethics Committee of the Medical Association of Westphalia-Lippe and the University of Münster (Approval number 2021-063-f-S). Written informed consent was obtained from all participants before blood sampling.

### 2.2. PBMCs Isolation

Peripheral blood mononuclear cells (PBMCs) were isolated by density gradient centrifugation. Venous blood was drawn from the antecubital vein via the aspiration method using S-Monovette (Sarstedt, Nümbrecht, Germany) containing 1.6 mg/mL of EDTA. The samples were mixed by gently inverting the tubes two to three times. The tubes were carefully transferred to the sterile bench for PBMCs isolation. In total, 20 mL of blood was gently mixed with 15 mL of PBS; a total of 35 mL of diluted blood was poured on top of 15 mL of Ficoll-Paque Plus (GE Healthcare, Braunschweig, Germany). The layers were separated by centrifugation at 400× *g* for 30 min at 20 °C without brake. Cells were carefully harvested from the interphase and subsequently washed twice with MACs buffer (PBS, 0.5% bovine serum albumin (BSA), 2 mM EDTA), then harvested from washing by centrifugation at 200× *g*, 10 min at 4 °C. Cells were then resuspended into 5 mL of PBS. The number of PBMCs was determined using an automated cell counter (TC20, BIO-RAD, Feldkirchen, Germany).

### 2.3. CD+14 Monocyte Isolation

From the pool of PBMCs, monocytes were isolated by positive selection with anti-human CD14 microbeads (Miltenyi Biotech, Bergisch Gladbach, Germany). PBMC suspensions in the range amount of 2 to 5 × 10^7^ cells were centrifuged at 300× *g*, 10 min at 4 °C to remove PBS, and the cell pellet was resuspended into 80 µL of cold MACs buffer containing 20 µL of CD14 microbeads. The mixture was incubated at 4 °C for 15 min. After incubation, 2 mL of cold MACs buffer was added, cells were harvested by centrifugation at 300× *g*, 10 min at 4 °C, and subsequently resuspended into 500 µL of cold MACs buffer. For magnetic separation, the S-sized column filled with ferromagnetic spheres, referred to as the ‘MS column’ (cat log # 130-042-201, Miltenyi Biotec), was fixed into a MACs separator (cat log # 130-042-102). The column matrix was equilibrated with 500 µL of MACs buffer before being loaded with the cell suspension. Unlabelled cells were passed through the column, while the CD14-positive cells were retained. Afterwards, the column was washed three times with 500 µL of cold MACs buffer to remove unspecific binding cells. To obtain the CD14^+^ monocytes, the column was removed from the separator and placed on a new 15 mL Falcon tube. The column was refilled with 1 mL of MACs buffer, and immediately, the magnetically labelled cells were flushed out by firmly pushing the plunger into the column. The number of CD+14 monocytes were counted and used for the further steps.

### 2.4. Pan Monocyte Isolation

The 1 × 10^7^ PBMCs were resuspended in 40 µL of cold MACs buffer, 10 µL of FcR blocking reagent, and 10 µL of biotin-antibody cocktail. The resulting sample was mixed well using a pipette and incubated for 5 min in the fridge at 4 °C. Subsequently, 30 µL of MACs buffer and 20 µL of anti-biotin microbeads were added into the cell suspension, which was continuously incubated for another 10 min at 4 °C. After incubation, the cell suspension was filled with 500 mL of cold MACs buffer. For magnetic separation, the MS column was fixed into the MACs separator. The column was then equilibrated with 500 µL of MACs buffer before 500 mL of cell suspension was applied. Unlabelled monocytes were allowed to pass through the column and collected into a new 15 mL Falcon tube.

### 2.5. Flow Cytometry Analysis

The purity of isolated cells was analysed by flow cytometry on a BD Accuri C6 (BD Biosciences, Heidelberg, Germany). The 10^6^ cells were resuspended in 100 µL of MACs buffer and stained with 2 µL of anti-human CD14-FITC, 2 µL of CD45-PE, and 2 µL of propidium iodide solution (Miltenyi Biotech, Bergisch Gladbach, Germany) for 30 min in the dark at 4 °C. Conjugation with CD14-FITC antibody binds to classical monocytes which present the CD14 antigen, while conjugation with CD45-PE antibody binds to all human cells of hematopoietic origin, with the exception of erythrocytes and platelets. Propidium iodide was used to exclude the dead cells out of the cell analysis. The cells were centrifuged at 300× *g* for 10 min and resuspended in 500 µL of PBS for flow cytometric determination of purity. The analysis was carried out with BD Accuri C6 software (BD Biosciences, Heidelberg, Germany).

### 2.6. Impairing Pre-Activation of Isolated Monocytes with the Use of a Cell-Repellent Culture Plate Surface

The 10^6^ CD14^+^ monocytes isolated by positive selection were resuspended into 2 mL of RPMI medium supplied with 10% heat-inactivated foetal calf serum (FCS), 1% L-glutamine, and 1% penicillin/streptomycin. They were incubated in 6- or 12-well CELLSTAR^®^ cell culture plate (Cat #657970 and #665970, respectively, Greiner Bio-One GmbH, Frickenhausen, Germany) at 37 °C with 5% CO_2_ supplement. This special plate with a cell-repellent surface prevents cell attachment. After 24 h of incubation, the cells were harvested by centrifugation at 300× *g* for 10 min and resuspended into new working RPMI medium at the concentration of 1 × 10^6^ cell/mL.

### 2.7. Stimulation with Lipopeptides

Synthetic Lpp P2C (Pam2CSK4), P3C (Pam3CSK4), and Lyso (PamCysPamCSK4), which mimic the lipid moiety structures of bacterial Lpp, were used for the cell stimulation. All these Lpp were chemically synthesized by EMC (Tübingen, Germany). After isolation, PBMCs and monocytes were resuspended in RPMI medium supplied with 10% heat-inactivated FCS, 1% L-glutamine, and 1% penicillin/streptomycin. For the stimulation assay, cells were seeded at a concentration of 10^5^ cells/100 µL/well into 96-well cell culture plates and subsequently supplied with 100 µL of medium containing P2C, P3C and Lyso to obtain a 200 µL volume with a final concentration of 100 ng/mL. In the non-stimulation wells, 10^5^ cells were supplied with 200 µL of medium alone. Cell cultures were incubated for 5 h to measure TNF-α and 24 h to measure IL-6. Supernatants were collected and stored at −20 °C until they were used for ELISA.

### 2.8. ELISA

Human cytokine secretion was determined in cellular supernatants using the Invitrogen eBioscience ELISA Set Go kits (Fisher scientific, Schwerte, Germany) for human TNF and IL-6 according to the manufacturer’s instructions. Briefly, Nunc-immuno 96-well plates (Cat #442404, Thermofisher, Germany) were coated with 100 µL of coating buffer containing capture antibody. After 1 h of blocking with diluent buffer, the plates were incubated with the sample supernatant at several dilutions (2, 5, and 10 times) for 2 h. Afterwards, the plates were incubated with 100 µL of diluent buffer containing detection antibody for 1 h and subsequently incubated for 30 min with 100 µL of diluent buffer containing streptavidin–HRP. Care was taken that the plates were washed 5 times with 300 µL of washing buffer (PBS, 0.05% Tween 20) using an ELISA washer in every incubation step. In the last step, 100 µL of TMB solution was added to develop colouring for 5 to 15 min incubation time. Thereafter, 50 µL of stop solution (2M H_2_SO_4_) was added to stop the development of the colour. The plates were measured at 450 nm with the Synergy HTX Multi-Mode reader (BioTek, Bad Friedrichshall, Germany).

### 2.9. Statistical Analysis

One-way analysis of variance (ANOVA) was employed to compare the differences between means. All of the statistical analyses were performed with GraphPad Prism. A *p*-value of >0.05 was considered ‘not significant’.

## 3. Results

### 3.1. Stimulation of Peripheral Blood Mononuclear Cells (PBMCs) with Bacterial Lipopeptides

PBMCs isolated by density gradient centrifugation were used for stimulation with different synthetic Lpps mimicking the bacterial Lpp structures. As a result, PBMCs that were stimulated with P2C, P3C, or Lyso peptides for 20 h produced a significant amount of TNF-α and IL-6, respectively, when compared to unstimulated cells (Figure 1A). Among these peptides, P3C triggered the highest cytokine TNF-α and IL-6 production. By FACS analysis, CD14^+^ monocytes were found to contribute to approximately 10% of the human cell population (Figure 1B; figures on the right are representative histograms which display results obtained with cells from one donor, both unstained and fluorescently stained with CD14-FITC and CD45-PE, respectively). The percentage of monocytes was calculated based on the number of CD14^+^ cells per the total CD45^+^ human cell.

### 3.2. Isolation of CD14^+^ Monocytes with CD14 Microbeads Cause Cell Activation without External Stimuli

For this preparation method, CD14^+^ cells labelled with CD14 microbeads were bound to the magnetic column while unlabelled cells passed in the void volume, allowing CD14^+^ monocytes to be eluted as a positively selected cell fraction. The average purity of monocytes was found to be above 90%, as determined by FACS (Figure 2A; the figures on the right are representative histograms which display results obtained with cells from one donor, both unstained and fluorescently stained with CD14-FITC and CD45-PE, respectively). The CD14^+^ monocytes were incubated with synthetic peptides used for PBMC stimulation for 5 h to measure TNF-α concentration and 20 h to measure IL-6 concentration. In the absence of external stimuli, the background cytokine release of CD14^+^-isolated cells was so high that no significant difference between unstimulated cells and cells stimulated with P2C or Lyso was detected (Figure 2B). Only P3C-stimulated CD14^+^ monocytes showed a significantly increased TNF-α amount (Figure 2B). With respect to stimulation with IL-6, both the unstimulated and stimulated cells secreted so much IL-6 that no difference could be observed between samples (Figure 2B).

### 3.3. Pan Monocyte Isolation Caused Cell Activation

For this preparation method, a cocktail of biotin-conjugated monoclonal antibodies directed against all antigens that are not expressed on human monocytes and anti-biotin labelled magnetic microbeads were used. Monocytes pass in the column void volume, while all other blood cells contained in the Ficoll interphase are retained in the magnetic column. This method was thought to preclude any monocyte pre-activation prior to exposure with external stimuli. The average purity of monocytes was found to be over 80%, as determined by FACS (Figure 3A; the figures on the right are representative histograms which display results obtained with cells from one donor, both unstained and fluorescently stained with CD14-FITC and CD45-PE, respectively). The pan monocytes were incubated with the different synthetic peptides used for the PBMC stimulation. In the absence of external stimuli, pan isolated monocytes revealed results similar to monocytes purified with the CD14^+^ procedure, since unstimulated cells secreted cytokines to an extent as high as upon P2C and Lyso stimulation (Figure 3B). Only P3C induced a significantly higher TNF-α and IL-6 amount in monocytes when compared to unstimulated cells (Figure 3B).

### 3.4. Incubating Monocytes with Cell-Repellent Surface Wells Allowed the Cells to Quieten Down after Purification

To allow the cells to quieten down following isolation using the CD14^+^ positive selection procedure, cells were allowed to rest in culture plates for 24 h prior to stimulation with peptides. Accordingly, the lipoprotein-induced activation profile was assessed in cells seeded either in normal cell culture plates or in wells equipped with a cell-repellent surface. Resting in normal cell culture plates did not reduce the pre-activation of the cells: the IL-6 amount produced in unstimulated cells was found to be as high as in peptide-stimulated cells (Figure 4A), while a lower TNF-α production in unstimulated cells and cells stimulated with peptides was observed (Figure 4A). In contrast, in the absence of Lpp, quiescent CD14^+^ monocytes produced neither detectable TNF-α nor IL-6, yet a significant and large amount of both TNF-α and IL-6 could be elicited in cells stimulated with the lipopeptides used for PBMC stimulation (Figure 4B).

## 4. Discussion

We presented a straightforward and efficient method to prepare pure, fully resting monocyte populations from PBMCs derived from whole human blood. The complete procedure is illustrated in Figure 5. By using staphylococcal Lpp as stimuli, we were able to show that these monocyte preparations, while completely non-activated prior to experimentation, were fully activatable and elicited a cytokine response, allowing us to discern the individual activation patterns of the various stimuli assessed.

It has been widely recognized that the purification process is critical for any subsequent in vitro analyses, and this challenge has prompted researchers to develop and communicate various methods, each carrying advantages and disadvantages, depending on considerations such as the purpose of subsequent usage, time, and cost. The hitherto most commonly employed method consists of monocyte isolation from lymphocytes by adherence [9,10,22]. However, this method was found to result in the secretion of high levels of TNF-α and IL-6 [23]. The addition of exogenous GM-CSF was found to significantly inhibit monocyte proliferation for up to 17 days [23]. Therefore, the use of an adherence-based monocyte preparation method could be appropriate for the purpose of macrophage generation, yet it may not be suitable for monocyte activation assays. Other cost-effective methods to isolate monocytes may employ cold aggregation or a Percoll gradient. The cold aggregation method, however, yielded a cell viability of approximately 95% yet only poor monocyte enrichment, while the Percoll gradient method allowed for higher monocyte purity yet only approximately 50% cell viability [24].

These adherence- or density-based methods have been largely amended with the employment of magnetic beads technologies, allowing for high purity and cell viability following separation/enrichment [24,25]. Compared to CD14^+^ monocyte positive selection, pan monocyte isolation was thought to cause less cell stress for isolated monocytes since the monocytes were not ‘selected’ by affinity to beads and subsequent retention in the magnetic column but directly released with the void volume. Recently, in a comparative study, isolation yields, purity, viability, and cell phenotype of monocytes (and monocyte-derived macrophages) were analysed with cells isolated by either of three methods: the magnetic-bead-supported, CD14^+^ positive and the pan (negative) monocyte isolation procedur

Es, as well as the standard plastic adhesion method [22]. As a result, it was found that monocyte preparations obtained with the positive selection method allowed for the largest yield, whereas preparations obtained with negative isolation were highly contaminated with platelets [22]. In this study, it was also demonstrated that different isolation techniques lead to a varying exposure of cell phenotype markers (such as CD80, CD163, HLA-DR, and CD206) with the consequence of a varying cell response to external stimuli [22]. In fact, our study revealed that both positive and negative selection lead to important cell activation owed to the purification procedure and precluding meaningful analyses of stimuli-triggered activation when using these non-resting monocyte preparations. However, the material used for cell incubation is also important; the pre-activation was dampened by using cell-repellent surface wells (Figure 4). Previous studies have demonstrated the impact of polymer surfaces on cell behaviours and host immune responses [26,27,28]. The adhesion of monocytes to plastic materials [29] such as polystyrene results in complex cellular events such as cell spreading, calcium release [30], and mRNA expression of interleukins [31] and other factors, i.e., cellular reactions critical for any in vivo assessment of cell activation. The introduction of cell-repellent-surface-modified culture plate materials now offers an elegant option to prevent the untoward effects of nonspecific activation. In addition to interaction with plastic surfaces, monocytes may be (pre-)activated by growth medium containing FCS. Hence, we inactivated putative, heat-sensitive, cell-activating factors contained in FCS by heat treatment; yet, it cannot be excluded that additional heat-resistant factors in FCS may still be functional. Our analyses, however, reveal that additional FCS factors triggering pre-activation may not be any more active following the monocyte quiescence step.

The limitation of this procedure is time consuming, with an extra 24 h incubation for cell resting. Moreover, monocytes harvested from cell-repellent surfaces should be transferred into fresh medium, and the number of cells should be counted due to a number of dead cells during the incubation (approximately 30% dead cells). Of note, we observed pre-activated monocytes isolated with CD14^+^ magnetic beads from fresh blood but not from buffy coats.

Resolving this dilemma, our study could now demonstrate that any pre-activation of isolated monocytes (due to CD14^+^ selection) could be largely impeded by allowing the cells to quieten down in cell-repellent surface plates for 24 h before stimulation, allowing for preparations of large yield and the purity of non-activated yet readily stimulus-responding cells. Hence, this method appears to be particularly valuable for the purification of monocytes from donor-derived fresh blood in order to analyse the individual activation pattern.

## Figures and Tables

**Figure 1 biomolecules-12-00395-f001:**
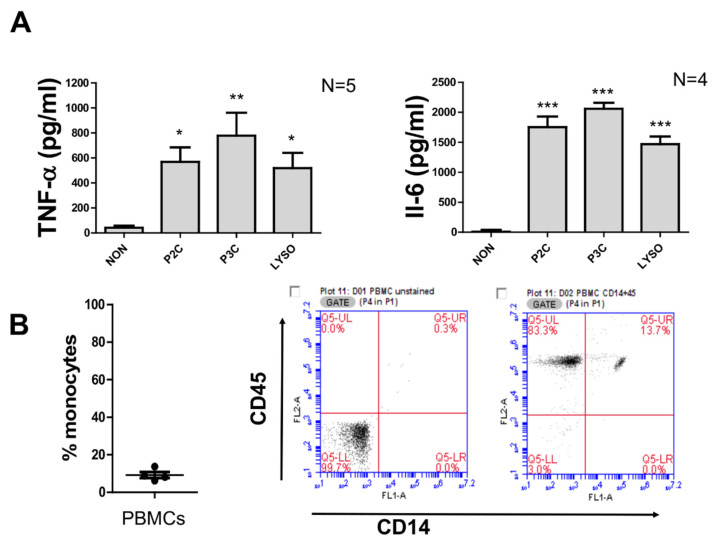
Cytokine production of PBMCs induced by lipopeptides. (**A**). Cytokine production of PBMCs with 100 ng/mL of P2C, P3C, or Lyso. The control samples (non) were used without adding any stimulators. TNF values were measured by ELISA from supernatant of 5 h stimulation, and IL6 and IL-10 were measured by ELISA from supernatant of 18 h stimulation. Samples from 5 donors were analysed in triplicate. Error bars represent the SEM. Statistical significances were calculated between the treated cells compared to control (non) by using Kuskal–Wallis test * *p*< 0.05, ** *p*< 0.01, *** *p*< 0.001. (**B**). Percentage of monocytes in isolated PBMCs. Cells were stained with CD14-FITC and CD45-PE antibodies. Data were obtained by flow cytometry from 4 donors. Flow cytometric analysis from a representative experiment with PBMCs unstained (left) and stained (right) with CD14-FITC and CD45-PE antibodies.

**Figure 2 biomolecules-12-00395-f002:**
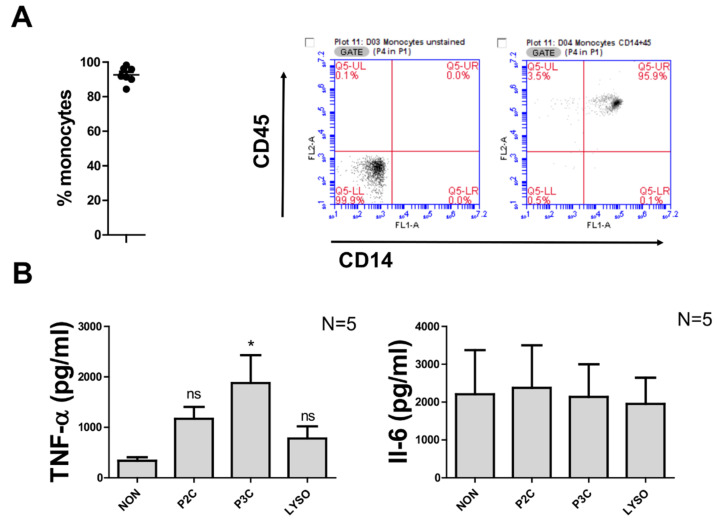
Monocyte production of CD14^+^ monocytes; positive isolation without resting. (**A**). Percentage of CD14^+^ monocytes from purified monocytes by CD14^+^ magnetic bead. Cells were stained with CD14-FITC and CD45-PE antibodies. Data were obtained from 7 donors. Flow cytometric analysis from a representative experiment with CD14^+^ monocytes unstained (left) and stained (right) with CD14-FITC and CD45-PE antibodies. (**B**) The 10^5^ monocytes were stimulated with 100 ng/mL of P2C, P3C, or Lyso. The control samples (non) were used without adding any stimulators. TNF values were measured by ELISA from supernatant of 5 h stimulation, and IL6 and IL-10 were measured by ELISA from supernatant of 18 h stimulation. Samples from 4 donors were analysed in triplicate. Error bars represent SEM. Statistical significances were calculated between the treated cells compared to control (non) by using Kruskal–Wallis test: * *p* < 0.05; ns, not significant.

**Figure 3 biomolecules-12-00395-f003:**
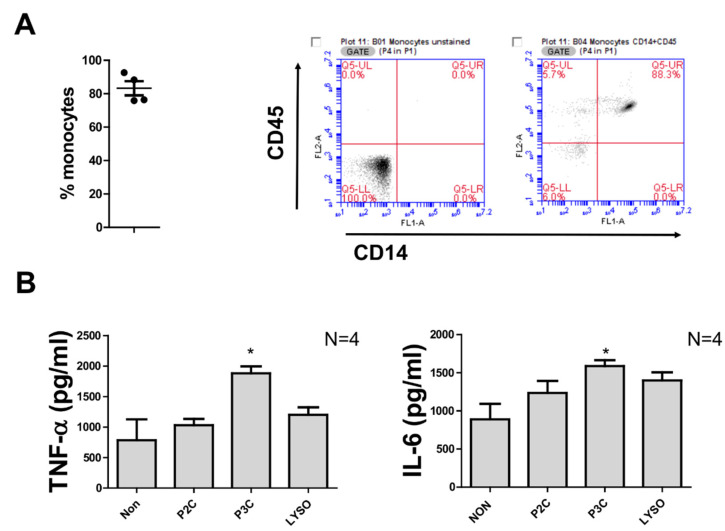
Cytokine production of pan monocytes isolation without resting. (**A**) Percentage CD14^+^ monocytes from purified pan monocytes. Cells were stained with CD14-FITC and CD45-PE antibodies. Data were obtained by flow cytometry from 4 donors. Flow cytometric analysis from a representative experiment with pan isolated monocytes unstained (left) and stained (right) with CD14-FITC and CD45-PE antibodies. (**B**) The 10^5^ monocytes were stimulated with 100 ng/mL of P2C, P3C, or Lyso. The control samples (non) were used without adding any stimulators. TNF values were measured by ELISA from supernatant of 5 h stimulation, and IL6 and IL-10 were measured by ELISA from supernatant of 18 h stimulation. Samples from 4 donors were analysed in triplicate. Error bars represent the SEM. Statistical significances were calculated between the treated cells compared to control (non) by using Kruskal–Wallis test: * *p* < 0.05.

**Figure 4 biomolecules-12-00395-f004:**
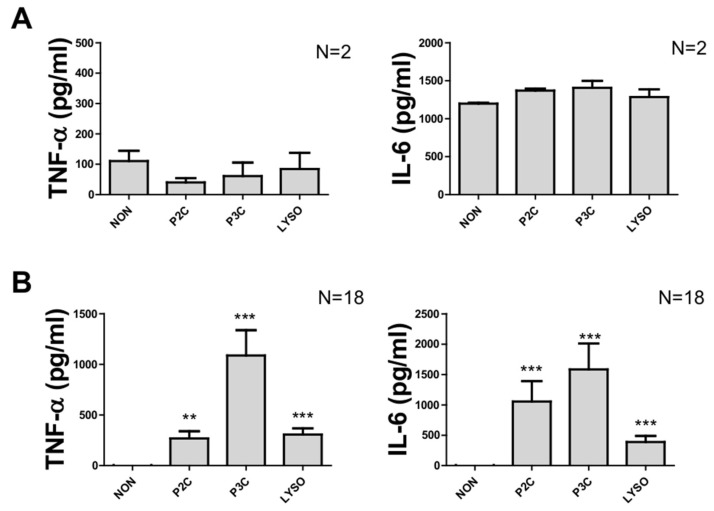
Cytokine production of CD14^+^ monocytes with resting. After 24 h resting in RPMI medium in normal cell culture plate (**A**) or in repellent surface plate (**B**), the monocytes were harvested and resuspended in a fresh RPMI medium at the concentration of 10^6^ cells/mL. The 10^5^ cells were stimulated with 100 ng/mL of P2C, P3C, or Lyso. The control samples (non) were used without adding any stimulators. TNF values were measured by ELISA from supernatant of 5 h stimulation, and IL6 and IL-10 were measured by ELISA from supernatant of 18 h stimulation. Samples from 2 donors (**A**) and 18 donors (**B**) were analysed in triplicate. Error bars represent the SEM. Statistical significances were calculated between the treated cells compared to control (non) by using Kruskal–Wallis test: ** *p* < 0.01, *** *p* < 0.001.

**Figure 5 biomolecules-12-00395-f005:**
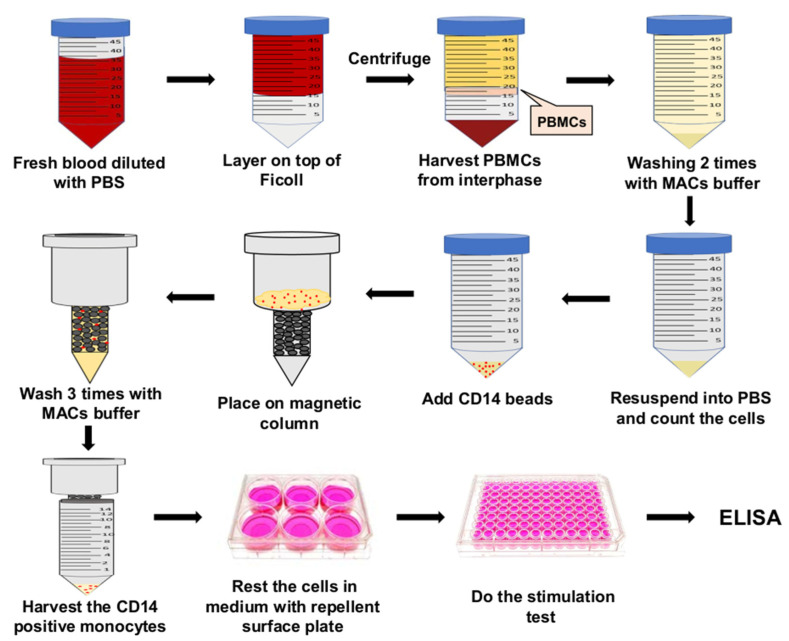
Flow chart of the human monocyte isolation procedure and stimulation with lipopeptides. Fresh blood from human volunteers is first processed over a Ficoll gradient. Monocytes are then purified from the PBMC layer by bead CD14^+^ affinity using magnetic separation technique and allowed to rest for 24 h in a repellent-surface-equipped plate. Subsequently, non-activated and pure monocyte populations are used for cytokine production upon lipoprotein stimulation. For further details, please refer to Materials and Methods.

## Data Availability

Not applicable.

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
