# Peer review of "Quiescence of Human Monocytes after Affinity Purification: A Novel Method Apt for Monocyte Stimulation Assays"

_biomolecules, 2022, doi:10.3390/biom12030395_

Round 1

Reviewer 1 Report

Nguyen et al., investigated whether isolated monocyte from PBMCs release pro-inflammatory cytokines TGF-a and IL-6 using conventional methods (e.g. magnetic cell isolation, pan monocyte selection) and further modify this procedure by adding a quiescence step to resolve this problem. For the application, this monocyte isolate method precluded cell activation when studying lipoproteins stimulation on cellular level. The manuscript is well-written. There are a couple of minor revisions before publication:

  1. flow cytometry figure quality needs to be improved.
  2. please explain the possible mechanism of polymer modified cell behaviors, which result in resting state of monocytes
  3. authors need to provide the cell number before and after incubation on the repellant plate.

Author Response

We thank the Reviewer for the overall positive evaluation of our paper. Our point-by-point disposition to the Reviewer’s critique is as follows:

There are a couple of minor revisions before publication:

  1. flow cytometry figure quality needs to be improved.

We have enhanced the resolution of flow cytometry figures (Fig 1-3)

  1. please explain the possible mechanism of polymer modified cell behaviors, which result in resting state of monocytes

As requested, we now discuss this issue in the discussion section of the manuscript (lines 327-332):

Adhesion of monocytes to plastic materials [29] such as polystyrene results in complex cellular events such as cell spreading, calcium release [30], and mRNA expression of interleukins [31] and other factors, i.e. cellular reactions critical for any in vivo assessment of cell activation. The introduction of cell-repellent surface modified culture plate materials now offers an elegant option to prevent the untoward effects of nonspecific activation.

  1. authors need to provide the cell number before and after incubation on the repellant plate.

We have added the information in the line 334: (approximately 30% dead cells)

Reviewer 2 Report

I recommend the following changes to the manuscript, in bold:

Introduction line 24: ¨Amounting to a proportion of approximately 10% of all blood leukocytes, monocytes ...¨

Introduction line 39: ¨... i) peripheral blood monocytes mononuclear cell (PBMCs) isolation...¨

Discussion line 318: ¨ ...Indeed, previous studies has have ...¨

In addition, I was wondering about the rational behind having 10% FCS in the culture medium, which may also contain factors that may activate monocytes. Would it thus be possible to use culture medium without any serum, i.e. serum-free medium? Please, comment and maybe add this point to the discussion part.

Author Response

Reviewer 2:

I recommend the following changes to the manuscript, in bold:

Introduction line 24: ¨Amounting to a proportion of approximately 10% of all blood leukocytes, monocytes ...¨

Introduction line 39: ¨... i) peripheral blood monocytes mononuclear cell (PBMCs) isolation...¨

Discussion line 318: ¨ ...Indeed, previous studies has have ...¨

all done as requested

In addition, I was wondering about the rational behind having 10% FCS in the culture medium, which may also contain factors that may activate monocytes. Please, comment and maybe add this point to the discussion part.

Would it thus be possible to use culture medium without any serum, i.e. serum-free medium?

It is our experience and widely assumed that most types of human cells will not grow properly in serum-free medium. However, as the Reviewer rightly asserts, FCS may contain factors priming or even activating monocytes. We tried to address this problem at least in part by using FCS that was inactivated at 56 °C for 1 h prior to use. Unfortunately, this information was not given in the initial manuscript, we now add it to the revised manuscript (line 133& 145).

Moreover, we introduced the following statement (Line 332-337):

In addition to interaction with plastic surfaces, monocytes may be (pre-)activated by the growth medium containing FCS. Hence, we inactivated putative heat-sensitive, cell activating factors contained in FCS by heat treatment, yet, it cannot be excluded that additional heat-resistant factors in FCS may still be functional. Our analyses, however, reveal that additional FCS factors triggering preactivation may not be anymore active following the monocyte quiescence step.